# Restoration of Hip Geometry after Femoral Neck Fracture: A Comparison of the Femoral Neck System (FNS) and the Dynamic Hip Screw (DHS)

**DOI:** 10.3390/life13102073

**Published:** 2023-10-17

**Authors:** Marcel Niemann, Tazio Maleitzke, Markus Jahn, Katharina Salmoukas, Karl F. Braun, Frank Graef, Ulrich Stöckle, Sebastian Meller

**Affiliations:** 1Charité—Universitätsmedizin Berlin, Corporate Member of Freie Universität Berlin, and Humboldt-Universität zu Berlin, Centre for Musculoskeletal Surgery, Augustenburger Platz 1, 13353 Berlin, Germany; tazio.maleitzke@charite.de (T.M.); markus.jahn@charite.de (M.J.); katharina.salmoukas@charite.de (K.S.); frank.graef@charite.de (F.G.); ulrich.stoeckle@charite.de (U.S.); sebastian.meller@charite.de (S.M.); 2Berlin Institute of Health at Charité—Universitätsmedizin Berlin, Julius Wolff Institute for Biomechanics and Musculoskeletal Regeneration, 13353 Berlin, Germany; 3Berlin Institute of Health at Charité—Universitätsmedizin Berlin, BIH Biomedical Innovation Academy, BIH Charité Clinician Scientist Programme, Anna-Louisa-Karsch-Straße 2, 10178 Berlin, Germany; 4Department of Trauma Surgery and Orthopaedics, BG Hospital Unfallkrankenhaus Berlin gGmbH, 12683 Berlin, Germany; 5Department of Trauma Surgery, University Hospital Rechts der Isar, Technical University of Munich, 81675 München, Germany; karl.braun@gmail.com

**Keywords:** dynamic hip screw, femoral neck system, femoral neck fracture, individual medicine, minimally invasive surgery, multiple trauma, geriatrics

## Abstract

Background: The femoral neck system (FNS) was introduced as a minimally invasive fixation device for managing femoral neck fractures. Objective: To compare radiographic, clinical, and patient-reported outcome measures (PROMs) of femoral neck fracture patients following FNS compared to dynamic hip screw (DHS) implantation combined with an anti-rotational screw. Methods: Patients who underwent closed reduction and internal fixation of a femoral neck fracture between 2020 and 2022 were retrospectively included. We measured leg length, femoral offset, and centrum–collum–diaphyseal (CCD) angle in plain radiographs. Scar length, Harris Hip Score, short-form health survey 36-item score (SF-36), and Numeric Rating Scale (NRS) were assessed during follow-up visits. Results: We included 43 patients (22 females) with a median age of 66 (IQR 57, 75). In both groups, leg length differences between the injured and the contralateral side increased, and femoral offset and CCD angle differences were maintained over time. FNS patients had shorter scars and reported fewer emotional problems and more energy. There were no differences between groups regarding the remaining SF-36 sub-scores, Harris Hip Score, and NRS. Conclusions: The FNS allows for a comparable leg length, femoral offset, and CCD angle reconstruction while achieving similarly high functional and global health scores to the DHS.

## 1. Introduction

In 2018, the femoral neck system (FNS) was introduced as a percutaneous system for the treatment of femoral neck fractures [1,2]. Osteosyntheses of the femoral neck require rotational stability and compression [3], which can be ensured by the design of the FNS [4]. Correspondingly, a biomechanical study has shown that the FNS is a reliable alternative to the commonly used dynamic hip screw (DHS) combined with an anti-rotational screw [5].

Clinical studies have shown that applying the FNS in femoral neck fracture treatment can improve functional outcomes [6,7,8,9,10,11] while reducing patients’ perioperative complication risk [1,2,12]. The most frequently observed complications following FNS treatment are a shortening of the femoral neck and the development of avascular necrosis of the femoral head [6,13,14]. Previous studies have reported severe pain in daily activities due to avascular necrosis development [15,16], which may lead to revision surgery and hip arthroplasty [6,9,17].

Though studies reporting patient-reported outcome measures (PROMs) following FNS treatment have been published, most compared the FNS to multiple cannulated screws [6,7,8,10,18,19]. Reported PROMs were mainly satisfactory in both groups, with the FNS obtaining superior results in most cases [6,7,8,9,10,11]. Thus far, data on PROMs comparing the DHS and the FNS have not been published.

We recently reported our first perioperative experiences of femoral neck fracture patients following FNS treatment compared to DHS treatment combined with an anti-rotational screw [20]. We observed a significantly shorter operating room time and a lower dose area product in the FNS group. There were no differences in length of stay or changes in perioperative hemoglobin concentration between groups.

Following the completion of regular outpatient follow-up visits, this study aims to assess radiographic and functional results and PROMs following femoral neck fracture treatment employing the DHS combined with an anti-rotational screw or the FNS. Further, general and surgical site complications were evaluated.

## 2. Methods

We examined our hospital’s electronic medical data system SAP (SAP ERP 6.0 EHP4, SAP AG, Walldorf, Germany) for femoral neck fracture patients treated between January 2020 and December 2022. Patients were included if the treatment was closed reduction and internal fixation using either the DHS (dynamic hip screw, DePuy Synthes, Warsaw, IN, USA) combined with an anti-rotational screw or the FNS (femoral neck system, DePuy Synthes). Inclusion was independent of fracture complexity and pre-existing medical conditions. Local institutional review board approval was obtained before study initiation (application number EA4/141/21).

The electronic medical data system SAP was used to perform a retrospective chart review for patients’ characteristics. These included age, gender, the American Society of Anesthesiologists (ASA) physical status classification system, Charlson Comorbidity Index [21], body mass index (BMI), mechanism of injury, and length of stay. Fracture morphology was classified according to Pauwels [22] and Garden [23].

Follow-up visits were regularly scheduled in our outpatient department six months after surgery. Besides the standard clinical investigation protocol and radiographic follow-up, we offered patients a more detailed analysis of their clinical function. For this, we used the Harris Hip Score [24], the short-form health survey 36-item score (SF-36) [25], and the Numeric Rating Scale (NRS), as long as patients had not been converted to a total hip arthroplasty. The Harris Hip Score and the SF-36 range from 0 to 100, with higher scores indicating better function or lower impairment. The NRS ranges from 0 to 10, with higher scores indicating more pain. Last, scar lengths were measured, and the rate of and reasons for revision surgeries were noted.

We assessed radiographic parameters in plain anterior–posterior radiographs of the pelvis using the MERLIN Diagnostic Workcenter (MERLIN Diagnostic Workcenter for Microsoft Windows, Version 5.8.1, Phönix-PACS GmbH, Freiburg im Breisgau, Germany). As previously described, Pauwels and centrum–collum–diaphyseal (CCD) angles were measured in the last preoperative and first postoperative radiographs [20]. Also, CCD angles, femoral offset, and leg length were evaluated using the earliest postoperative and follow-up radiographs at six months (Figure 1).

Statistical analysis was performed using GraphPad Prism (GraphPad Prism 10 for macOS, Version 10.0.2, GraphPad Holdings, LLC, San Diego, CA, USA). Data distribution was assessed using histograms and Q–Q plots. For discrete and continuous variables, the Mann–Whitney *U* test was used for independent samples, and the Wilcoxon signed-rank test for dependent samples. Comparisons between healthy and injured extremities and repetitive measurements within the same extremity were defined as dependent measures. For categorical variables, Fisher’s exact test was used. We performed outlier detection in indicated analyses using the ROUT method with Q = 0.1% [26]. Unless stated otherwise, discrete and continuous variables are represented as the median and interquartile range (IQR), and categorical variables as frequencies and portions of a whole (%). All *p*-values are two-tailed, and *p*-values < 0.05 were considered statistically significant.

## 3. Results

### 3.1. Demographics

Between January 2020 and December 2022, 46 patients (22 (47.83%) female) with a median age of 66 years (IQR 57, 75) were treated due to a femoral neck fracture at our clinic. Most patients (35 (76.09%)) reported low-impact injuries (falls from standing height). Six patients (13.04%) had bicycle accidents, three (6.52%) had motorized scooter accidents, one (2.17%) had an inline skate accident, and one (2.17%) patient was involved in a car accident. Of these, 23 patients (50.0%) received the DHS combined with an anti-rotational screw, and 23 (50.0%) received the FNS. Most DHS patients (22/23 (95.65%)) received a two-hole blade, and one patient (4.35%) received a four-hole blade due to the surgeon’s intraoperative decision. All FNS patients received a one-hole blade. Median ASA and Charlson Comorbidity Index were 2 (IQR 2, 3) and 3 (IQR 1, 5), respectively. Due to the expected perioperative risk, three patients (13.04%) of the DHS group and six patients of the FNS group (26.09%) received an in situ fixation. There were no differences regarding age, gender, ASA, Charlson Comorbidity Index, BMI, or fracture morphology distribution between groups.

### 3.2. Follow-Up

The scheduled follow-up visits were completed by 32 (69.57%) of the patients, which included 18 (78.26%) in the DHS group and 14 (60.87%) in the FNS group.

### 3.3. Radiographic Outcomes

Radiographic outcomes were assessed using the postoperative radiograph at 6-month follow-up (last follow-up) compared to the earliest postoperative radiograph (first follow-up).

In DHS patients, leg length did not differ at the first follow-up (injured: 69.80 mm [IQR 64.13, 74.54]; healthy: 72.99 mm [IQR 68.09, 78.02]; *p* = 0.12), but did at the last follow-up (injured: 69.15 mm [IQR 64.47, 71.76]; healthy: 73.01 mm [IQR 68.01, 79.94]; *p* < 0.01) (Figure 2a,b). In FNS patients, leg length differed at the first (injured: 67.53 mm [IQR 62.53, 72.49]; healthy: 69.44 mm [IQR 66.48, 75.67]; *p* < 0.01) and last follow-up (injured: 66.06 mm [IQR 60.70, 74.14]; healthy: 72.02 mm [IQR 67.62, 80.80]; *p* < 0.01) (Figure 2c,d). Leg length differences increased over time in both groups (DHS: −1.94 mm [IQR −4.98, 1.60] to −5.02 mm [IQR −10.88, −1.70], *p* < 0.01; FNS: −2.48 mm [IQR −7.46, 0.12] to −5.40 mm [IQR −12.57, −0.38], *p* = 0.05).

In DHS patients, femoral offset of the injured side was lower at first (injured: 37.72 mm [IQR 31.44, 41.34]; healthy: 46.74 mm [IQR 41.15, 51.69]; *p* < 0.01) and last follow-up (healthy: 38.69 mm [IQR 36.55, 41.56]; injured: 51.08 mm [IQR 41.05, 53.34]; *p* < 0.01) (Figure 2e,f). In FNS patients, femoral offset of the injured side was also lower at first (injured: 32.42 mm [IQR 29.33, 37.10]; healthy: 39.41 mm [IQR 36.10, 44.09]; *p* < 0.01) and last follow-up (injured: 35.21 mm [IQR 33.05, 41.36]; healthy: 39.65 mm [IQR 36.68, 46.21]; *p* < 0.01) (Figure 2g,h). Differences in femoral offset were stable over time in both groups (DHS: −9.60 mm [IQR −14.97, −4.86] to −12.27 mm [IQR −13.19, −3.88], *p* = 0.89; FNS: −6.86 mm [IQR −9.62, −2.73] to −5.63 mm [IQR −7.64, −1.46], *p* = 0.27).

DHS patients had more valgus initially (injured: 135.41° [IQR 127.35, 139.92]; healthy: 130.48° [IQR 127.00, 133.92]; *p* = 0.03), which was not observed at the last follow-up (injured: 132.54° [IQR 125.28, 138.77]; healthy: 129.11° [IQR 122.28, 133.94]; *p* = 0.39) (Figure 2i,j). Similarly, FNS patients had more valgus initially (injured: 137.60° [IQR 131.15, 141.61]; healthy: 132.64° [IQR 126.86, 135.82]; *p* = 0.02), which was not observed at the last follow-up (injured: 134.94° [IQR 127.72, 138.43]; healthy: 131.75° [IQR 126.39, 137.96]; *p* = 0.35) (Figure 2k,l). Over time, CCD angle differences stabilized in both groups (DHS: 3.33° [IQR −2.67, 8.53] to 0.06° [IQR −2.71, 7.77], *p* = 0.15; FNS: 4.96° [IQR −1.07, 8.56] to 2.70° [IQR −2.23, 5.90], *p* = 0.06).

As depicted in Figure 1, radiographic assessments were conducted in a standardized fashion using MERLIN Diagnostic Workcenter. Figure 3 provides an illustrative representation of the respective measurements for femoral offset, the CCD angle, and the radiographic leg length associated with each incorporated implant. The outcomes of each evaluation were directly annotated by the software within the radiograph.

### 3.4. Scar Lengths and PROM Assessment

The median scar length was 9.5 cm (IQR 8.5, 10.0) in the DHS group and 4.5 cm (IQR 4.0, 7.0) in the FNS group at the 6-month follow-up visit (*p* < 0.01) (Figure 4).

At the follow-up, the FNS patients reported significantly better emotional satisfaction and significantly more energy/less fatigue than DHS patients. There were no further significant differences between groups for Harris Hip Score, the remaining sub-domains of the SF-36, or NRS. See Table 1 for detailed results and comparisons between groups.

### 3.5. Revision and Mortality Rate

Eleven patients (47.83%) in the DHS group and two (8.70%) in the FNS group required revision surgery after a median follow-up of 31.00 (IQR 3.00, 73.00) and 34.00 weeks (IQR 10.00, 58.00), respectively. In the DHS group, reasons for revision were implant-related pain (N = 3 (27.27%)), femoral neck shortening (N = 3 (27.27%)), avascular necrosis (N = 1 (9.09%)), impaired wound healing (N = 2 (18.18%)), persistent local hematoma (N = 1 (9.09%)), and peri-implant fracture (N = 1 (9.09%)). The latter was a patient who fell from a standing height three weeks post surgery with no previous discomfort. In the FNS group, reasons for revision surgery were implant-related pain (N = 1 (50.00%)) and femoral neck shortening (N = 1 (50.00%)). No further revision surgeries were observed. There were no implant failures, including breaking or cut-out, in either of the groups.

### 3.6. Management of Complications

Patients with femoral neck shortenings and avascular necrosis underwent conversion total hip arthroplasty. Impaired wound healing was treated with local wound debridement and implant retention, local hematomas were treated with surgical evacuation, the peri-implant fracture received a conversion proximal femoral nail, and implants solely causing pain were removed.

One patient (4.35%) died 25 months after implantation of a DHS due to urosepsis. Another patient (4.35%) died nine months after FNS implantation from natural causes. Neither event was considered an adverse event related to the previous fracture management. The mortality rate did not differ between groups (*p* = 1).

Only a few patients had complications when treating their femoral neck fracture with a DHS with an anti-rotational screw. For example, a 57-year-old female patient was transferred to our hospital following a fall from standing height. She had a Pauwels type 3, Garden type 4 fracture (Figure 5a) and was treated with a DHS with an anti-rotational screw (Figure 5b). Despite an initially satisfactory follow-up, she presented with an avascular necrosis of the femoral head six months following surgery (Figure 5c), which required conversion to total hip arthroplasty. However, most patients treated with a DHS with an anti-rotational screw had a satisfactory outcome. For instance, a 40-year-old male patient also presented with a Pauwels type 3, Garden type 4 fracture (Figure 5d) and received a DHS with an anti-rotational screw (Figure 5e). The follow-up result was inconspicuous, and the fracture healed (Figure 5f).

A minority of patients had complications when femoral neck fractures were treated with an FNS. A 67-year-old male patient was transferred to our hospital following a fall from standing height. He had a Pauwels type 2, Garden type 3 fracture (Figure 6a) and was treated with an FNS (Figure 6b). Despite an initially satisfactory follow-up, he presented with a consecutive cutting-out of the implant at six months following surgery (Figure 6c). However, most patients treated with an FNS had a satisfactory outcome. For instance, a 62-year-old male patient presented with a Pauwels type 2, Garden type 2 fracture (Figure 6d) and received an FNS (Figure 6e). The follow-up result was inconspicuous, and the fracture healed (Figure 6f).

## 4. Discussion

This study assessed radiographic and functional follow-up data of patients treated for a femoral neck fracture using one of two minimally invasive systems—DHS or FNS. At the 6-month follow-up, radiographic leg length and femoral offset were significantly reduced on the previously injured side compared to the contralateral healthy side in both groups. Further, patients in both groups tended to have more varus in the CCD angle at follow-up. Still, patients who received the FNS had significantly shorter scars, fewer emotional problems, and more energy than those receiving the DHS. This has, thus far, not been described in the literature.

With globally rising life expectancies [27], the incidence of femoral neck fracture is similarly expected to keep increasing [28]. These injuries commonly constitute a life-changing event that frequently ends a previously independent lifestyle [29], as most patients do not reach pre-injury function and health-related quality of life [30]. There is a strong need for straightforward and standardized treatment concepts [31,32] to avoid fracture- and surgery-related adverse events [33]. In the case of femoral neck fracture, treating physicians must decide between osteosynthesis (FNS and DHS) and joint replacement surgery using hemi- or total arthroplasty. Especially for fracture patterns with presumably intact femoral head perfusion and fractures in patients for whom, due to comorbidities or concomitant injuries, a comparably more invasive arthroplasty is not appropriate [34].

Since its introduction, the FNS has added to the spectrum of osteosynthesis systems for femoral neck fracture management. These previously included multiple cannulated screws and the DHS. The additional anti-rotational screw accompanies the latter for increased rotational stability [5]. Within the last year, several meta-analyses have been published that compared the FNS and multiple cannulated screws [35,36,37,38,39]. The studies found less femoral neck shortening [36,38,39] and higher postoperative Harris Hip Score values [37,38,39] in patients treated with the FNS. However, most of the analyses were solely based on retrospective studies, and heterogeneity was high between studies.

Several single-center studies have provided data regarding applying the FNS in femoral neck fracture treatment [6,7,8,10,18,19,40,41,42]. In our previous analysis [20], we discussed Stassen et al.’s study, which reported an incision length of 45.3 ± 8.8 mm [40], which matches the observed scar length in our data set. They noted the following complications in thirty-four patients treated with the FNS: eight patients presented with persisting pain within the first year after surgery; four patients suffered from an avascular necrosis; two suffered from a cut-out of the blade; and two suffered from implant-related pain, which was resolved following implant removal [40]. Another study observed a revision rate of 8% within 18 months after surgery due to avascular necrosis, cut-out of the blade, or fracture non-union [41]. Stoffel et al. recently provided follow-up data on 125 femoral neck fracture patients treated with the FNS [42]. Patients’ age was similar to our cohort’s median age, as was the operating room time. However, most patients showed fewer comorbidities (mean Charlson Comorbidity Index of 1) and a different distribution of fracture complexity according to Garden (41.6%, 28%, 28%, and 2.4% of Garden I, II, III, and IV, respectively) compared to our cohort (4.35%, 73.91%, 17.39%, and 4.35% of Garden I, II, III, and IV, respectively). Revision rates of 6.4% at three months and 8.8% at 12 months were reported, mainly due to cut-out of the blade (1.6%), implant failure (1.6%), loss of reduction (1.6%), implant telescoping (0.8%), and non-union (0.8%) [42]. Compared to these studies, the reported revision rates of our cohort were higher, as we also considered implant removals due to implant-related pain as complications.

Studies comparing the application of the FNS and the DHS for the management of femoral neck fracture reported a reduction in intraoperative fluoroscopy [20,43], shorter operating room time [9,20,43,44], and shorter length of stay [43,44] when using the FNS. The authors did not observe any differences in revision rates between the two systems [9,44], proving the FNS to be a competitive system compared to the DHS. Further, the cut-out rate of the blade was comparable to that of the DHS system (12.4% vs. 10.2%) [44]. This observation is supported by a recent biomechanical analysis that observed no differences regarding axial stiffness, failure load, varus collapse, or axial screw migration between FNS and DHS [45]. Thus far, poor blade positioning is the only significant predictor for a cut-out of the blade [44]. We did not observe any implant failure or cut-out in our cohort.

Two of the most common adverse events following osteosynthesis of a femoral neck fracture are femoral neck shortening and avascular necrosis development, as patients frequently need conversion arthroplasty [13,14]. Schuetze et al. did not observe statistically significant differences in femoral neck shortening between the two systems over a follow-up period of 13 months (FNS: 5.3 ± 1.9 mm; DHS: 4.8 ± 2.1 mm) [45]. Similarly, Vazquez et al. reported 5.6 ± 5.9 mm vs. 5 ± 6.7 mm shortening relative to the femoral neck length and 4.9 ± 7.9 mm vs. 6.7 ± 4.0 mm shortening relative to the acetabular roof over six months when comparing the DHS and FNS, respectively [9]. In the present study, we compared leg lengths, femoral offset, and CCD angles to those on the contralateral site. This decision was made because osteosyntheses usually aim to restore the pre-injury anatomy, which is commonly assumed to some extent to match the contralateral side. In our study, leg length was more likely to be reconstructed in the DHS than in the FNS group. Leg length decreased in both groups over time, leading to rising length differences between injured and contralateral sides. Differences in femoral offsets were stable over the study period but lowered on the injured compared to the contralateral healthy site in both groups. CCD angles had more valgus after implantation of either an FNS or a DHS, an effect which then decreased over time. Previous articles have not reported femoral offsets and CCD angles comparing the DHS and FNS.

Stoffel et al. examined patients who had received the FNS. PROMs were overall satisfactory at 12 months post surgery, even though they were lower than the retrospectively assessed pre-injury scores [42]. In our analysis, the Harris Hip Score was comparably high in both groups. Contrary to Stoffel et al., we did not use the EQ-5D-5L Index but all domains of the SF-36. Patients who received an FNS rarely reported emotional problems but did report more energy than those who received a DHS with an anti-rotational screw. Despite no further statistically significant differences, most sub-scores of the SF-36 were higher in FNS patients. Also, both groups had low overall pain. The Harris Hip Score was also highly comparable to that reported by previous authors comparing the FNS and multiple cannulated screws [6,8,10,18,19]. In summary, the FNS provides highly satisfactory physical function and general health scores, which are not inferior to the results obtained when using the DHS.

Our study has a few limitations. First, the study cohort was small. More patients per examined group would have improved the power for statistical analyses. Authors have recently begun publishing their experiences using the FNS [6,7,8,10,18,19] but have primarily compared it to multiple cannulated screws. Accordingly, comparative data on the usage of the DHS and the FNS are still limited, so we compared follow-up data, including laboratory data, clinical data, radiographic measures, and PROMs. Second, the follow-up period was limited. The FNS was introduced in 2018 but first used in our clinic in 2021. This limits the follow-up period achievable up until now. We collected all available data, but some patients still missed follow-up visits. Future studies should aim to prolong follow-up periods to compare both systems for long-term outcome measures and complications. Third, we had a low follow-up rate regarding the PROMs assessment and, to a lower extent, the radiographic follow-up. Still, we observed statistical differences between some of the assessed outcome measures. However, the absence of significant differences concerning the remaining measures in a few patients might not necessarily lead to a judgment of equality between the examined groups. Larger cohorts might make it possible to detect relevant but minor differences between groups in the future. Last, we did not set age limits for study inclusion. However, most of our patients were rather old. Therefore, we cannot conclude any observations on young people with femoral neck fractures. Different age groups typically require different treatment strategies due to highly varying treatment goals. Future studies might need to focus on younger patients as well in order to conclude the optimal treatment strategy for these patients, as they represent the majority of multiple-injured patients. These highly volatile patients require individualized treatment concepts. Further, there were more highly unstable Garden type IV fractures in the DHS group. These highly unstable fractures may have a higher risk when managed using an FNS, but sufficient data to allow us to recommend a treatment strategy is, thus far, lacking. Yet, we have provided highly relevant outcome data and assessed several important factors following osteosynthesis treatment of the proximal femur. As previously described [20], we had no strict exclusion criteria concerning fracture complexity or patients’ comorbidities, which may have added to the ‘noise’ and should be considered in future study plans.

## 5. Conclusions

The FNS is a highly efficient and minimally invasive system for the management of femoral neck fractures. Compared to the DHS, the FNS allows for a comparable leg length, offset, and CCD angle reconstruction while achieving high functional and global health scores.

## Figures and Tables

**Figure 1 life-13-02073-f001:**
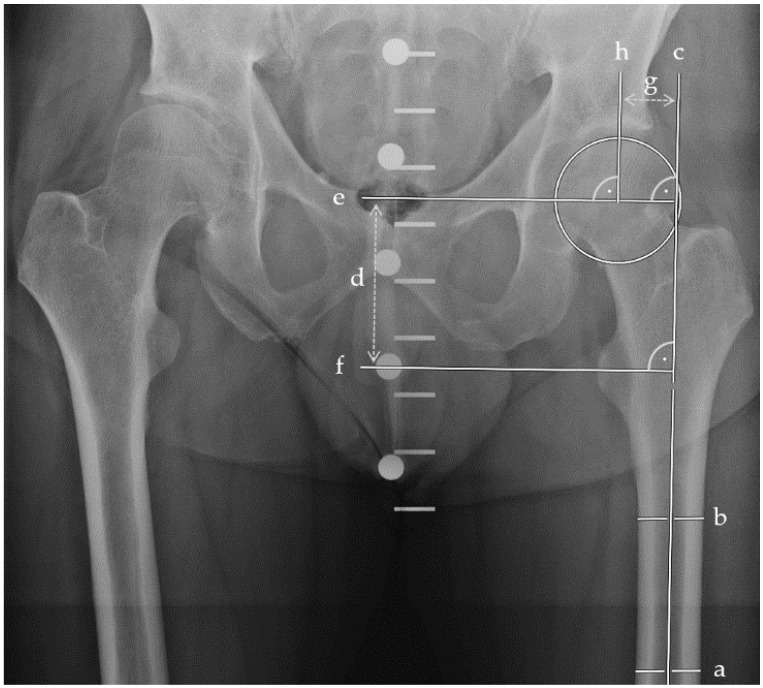
Plain anterior–posterior radiographs of the pelvis visualizing leg length and femoral offset measurements. Two parallel bisections of the femoral shaft (a, b) determined the longitudinal femoral shaft axis (c). Perpendicular to (c), the leg length (d) was measured between the femoral head centre (e) and the tip of the lesser trochanter (f). The femoral offset (g) was measured between the femoral head center (h) perpendicular to (c).

**Figure 2 life-13-02073-f002:**
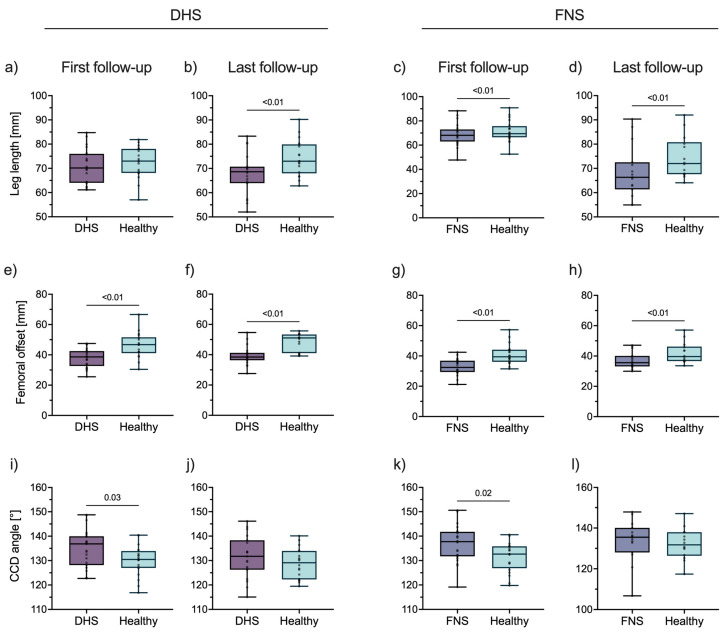
Radiographic comparison of injured and healthy sides. Leg lengths of DHS patients at (**a**) first and (**b**) last postoperative follow-up, and FNS patients at (**c**) first and (**d**) last postoperative follow-up. Femoral offset of DHS patients at (**e**) first and (**f**) last postoperative follow-up, and FNS patients at (**g**) first and (**h**) last postoperative follow-up. CCD angle of DHS patients at (**i**) first and (**j**) last postoperative follow-up, and FNS patients at (**k**) first and (**l**) last postoperative follow-up. Abbreviations: DHS, dynamic hip screw; FNS, femoral neck system; CCD, centrum–collum–diaphyseal.

**Figure 3 life-13-02073-f003:**
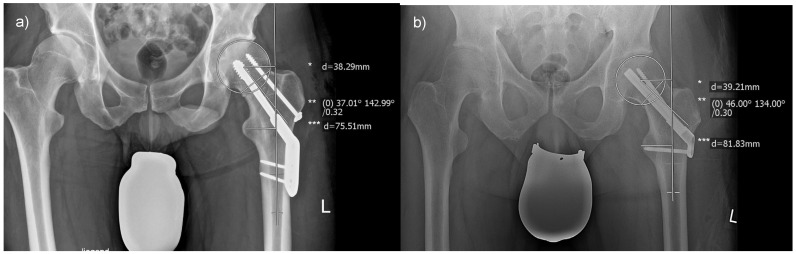
Measurements of radiographic outcome measures: (**a**) displays the measurement of a femoral neck fracture treated with a DHS, and (**b**) with an FNS. Abbreviations: *, femoral offset; **, centrum–collum–diaphyseal angle; ***, leg length.

**Figure 4 life-13-02073-f004:**
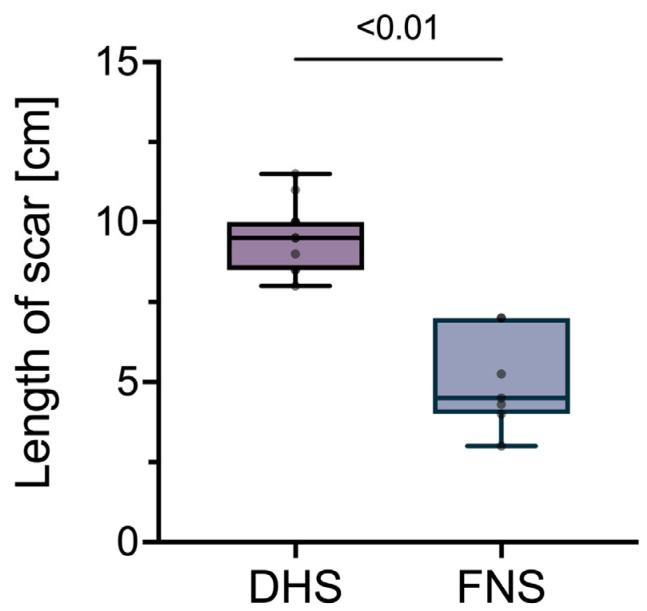
Comparison of scar lengths between DHS and FNS groups. While DHS patients had a median scar length of 9.5 cm (IQR 8.5, 10.0), FNS patients had a significantly shorter (*p* < 0.01) median scar length of 4.5 cm (IQR 4.0, 7.0). Abbreviations: DHS, dynamic hip screw; FNS, femoral neck system.

**Figure 5 life-13-02073-f005:**
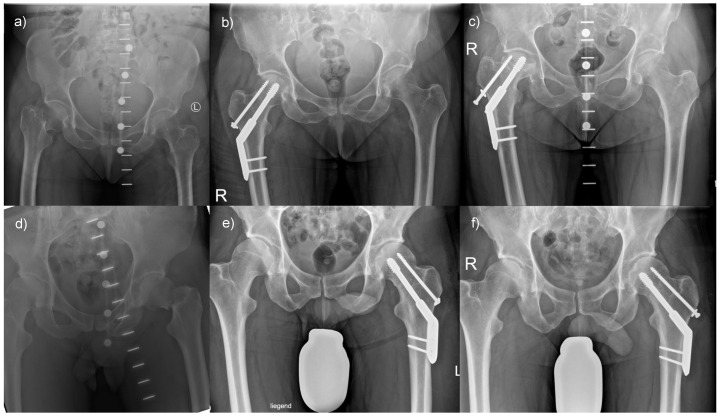
Outcomes of femoral neck fractures when treated with a DHS with anti-rotational screw: (**a**) A 57-year-old female patient presented with a highly unstable femoral neck fracture, (**b**) which was treated with a DHS with an anti-rotational screw, but (**c**) developed an avascular necrosis of her hip. In comparison, (**d**) a 40-year-old male patient presented with a comparable fracture morphology, (**e**) underwent analogous treatment, and (**f**) demonstrated favorable postoperative results.

**Figure 6 life-13-02073-f006:**
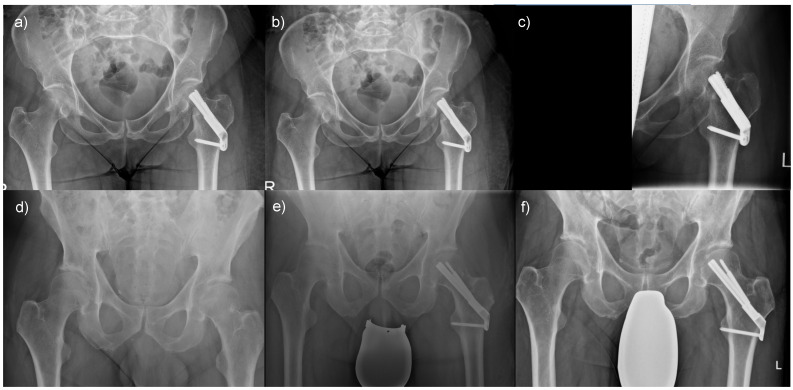
Outcomes of femoral neck fractures when treated with an FNS: (**a**) A 67-year-old male patient presented with an unstable femoral neck fracture, (**b**) which was treated with an FNS, but (**c**) presented with implant cut-out following surgery. In comparison, (**d**) a 62-year-old male patient presented with a comparable fracture morphology, (**e**) was treated in the same manner, and (**f**) had a favorable postoperative outcome.

**Table 1 life-13-02073-t001:** PROMs following FNS compared to DHS in combination with an anti-rotational screw.

		DHS (N = 14)	FNS (N = 11)	Statistics
Harris Hip Score		95.90 (IQR 72.20, 99.85)	94.70 (IQR 90.55, 99.85)	*p* = 0.53
SF-36	Physical functioning [%]	57.50 (IQR 21.25, 95.00)	75.00 (IQR 50.00, 90.00)	*p* = 0.41
	Role limitations due to physical health [%]	50.00 (IQR 0.00, 100.00)	100.00 (IQR 25.00, 100.00)	*p* = 0.19
	Role limitations due to emotional problems [%]	83.35 (IQR 0.00, 100.00)	100.00 (IQR 100.00, 100.00)	*p* = 0.02
	Energy/fatigue [%]	47.50 (IQR 38.75, 71.25)	55.00 (IQR 50.00, 80.00)	*p* = 0.04
	Emotional well-being [%]	58.00 (IQR 38.00, 89.00)	76.00 (IQR 60.00, 96.00)	*p* = 0.12
	Social functioning [%]	81.25 (IQR 31.25, 100.00)	87.50 (IQR 75.00, 100.00)	*p* = 0.42
	Pain [%]	62.50 (IQR 39.38, 92.50)	67.50 (IQR 55.00, 90.00)	*p* = 0.39
	General health [%]	62.50 (IQR 42.50, 78.75)	75.00 (IQR 50.00, 90.00)	*p* = 0.25
	Health change [%]	50.00 (IQR 43.75, 81.25)	50.00 (IQR 25.00, 100.00)	*p* = 0.70
NRS		2 (IQR 1, 5)	2 (IQR 1, 3)	*p* = 0.67

Abbreviations: DHS, dynamic hip screw; FNS, femoral neck system; SF-36, short-form health survey 36-item score; NRS, Numeric Rating Scale.

## Data Availability

The data presented in this study are available on request from the corresponding author. The data are not publicly available due to local institutional ethics board regulations.

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
