# Peer review of "Restoration of Hip Geometry after Femoral Neck Fracture: A Comparison of the Femoral Neck System (FNS) and the Dynamic Hip Screw (DHS)"

_life, 2023, doi:10.3390/life13102073_

Round 1

Reviewer 1 Report

This is a very well-written paper that provides insightful results to a surgery procedure that is on the increase due to greater life expectancies. I found that there were a lot of abbreviations which interrupted the ability to read the manuscript with a good flow. I wonder whether some of the abbreviations could be removed. Obviously, keep the abbreviations that are frequently used throughout the manuscript.

The quality of the English language is very good. 

Author Response

Dear Reviewer 1,

thank you very much for reviewing our manuscript.

This is a very well-written paper that provides insightful results to a surgery procedure that is on the increase due to greater life expectancies. I found that there were a lot of abbreviations which interrupted the ability to read the manuscript with a good flow. I wonder whether some of the abbreviations could be removed. Obviously, keep the abbreviations that are frequently used throughout the manuscript.

Answer: Thank you very much. We eliminated less commonly used abbreviations in the revised manuscript.

Yours sincerely

The corresponding author, on behalf of all authors

Reviewer 2 Report

I would suggest 2 improvements . One in limitations you must mention the limited number of patients in follow up as it’s very low. I understand that there are some statistical differences but with this few pts real differences in outcome are hard to delineate . Please show us an example of your post op images with measurement showing one each of a Dhs and FNS with marks for measurement included . I think that would be best and maybe pick a good and bad outcome for each. This is an online journal so images are very helpful and not limited . 

Author Response

Dear Reviewer 2,

thank you very much for reviewing our manuscript. This was relevant input to improve our manuscript.

I would suggest 2 improvements . One in limitations you must mention the limited number of patients in follow up as it's very low. I understand that there are some statistical differences but with this few pts real differences in outcome are hard to delineate.

Answer: We added this to the limitations of the revised manuscript.

Please show us an example of your post op images with measurement showing one each of a Dhs and FNS with marks for measurement included . I think that would be best and maybe pick a good and bad outcome for each. This is an online journal so images are very helpful and not limited.

Answer: We have included exemplary images of both good and bad outcomes for each implant. Further, we added a figure showing exemplary measurements of each implant.

Yours sincerely

The corresponding author, on behalf of all authors

Reviewer 3 Report

The authors of this article established the femoral neck system (FNS) as a minimally invasive fixation device for treating femoral neck fractures (FNF), with the goal of comparing various outcome metrics between FNS and dynamic hip screw (DHS) implantation with an anti-rotational screw (ARS). The research involved 43 patients, and both groups had increasing leg length differences, with FNS patients having shorter scars, less mental issues, and higher energy levels. However, no significant variations in other SF-36 sub-scores, HHS, or NRS were seen between groups, suggesting that FNS and DHS produce equivalent outcomes in terms of leg length, femoral offset, CCD angle repair, and functional and global health scores. Here are my comments:

Please comment on whether age factors or fracture type let to variation within the patient population of FNS and DHS?

Please comment if the authors observed changes in the bone mineral density in the femoral neck if each of these treatment affected the bone mineral density?

Author Response

Dear Reviewer 3,

thank you very much for reviewing our manuscript. This was relevant input in order to improve our manuscript.

The authors of this article established the femoral neck system (FNS) as a minimally invasive fixation device for treating femoral neck fractures (FNF), with the goal of comparing various outcome metrics between FNS and dynamic hip screw (DHS) implantation with an anti-rotational screw (ARS). The research involved 43 patients, and both groups had increasing leg length differences, with FNS patients having shorter scars, less mental issues, and higher energy levels. However, no significant variations in other SF-36 sub-scores, HHS, or NRS were seen between groups, suggesting that FNS and DHS produce equivalent outcomes in terms of leg length, femoral offset, CCD angle repair, and functional and global health scores. Here are my comments:

Please comment on whether age factors or fracture type let to variation within the patient population of FNS and DHS?

Answer: There were no differences between the age or fracture type distributions between both implants in the presented cohort. This was covered in the last sentence of the first paragraph of the results of the revised manuscript. Further, we discussed the influence of patients' age in the limitations in lines 349-352 and fracture type in lines 352-354.

Please comment if the authors observed changes in the bone mineral density in the femoral neck if each of these treatment affected the bone mineral density?

Answer: We did not regularly check bone mineral density in the follow-up of our patients. Concerning the plain radiographs, there were no apparent differences between implants concerning the osseous structures. However, this may not be sufficient to draw conclusions about the bone mineral density.

Yours sincerely

The corresponding author, on behalf of all authors

